# Expression of CD38 in Mast Cells: Cytological and Histotopographic Features

**DOI:** 10.3390/cells10102511

**Published:** 2021-09-22

**Authors:** Dmitri Atiakshin, Vera Samoilova, Igor Buchwalow, Markus Tiemann

**Affiliations:** 1Research and Educational Resource Center for Immunophenotyping, Digital Spatial Profiling and Ultrastructural Analysis Innovative Technologies, Peoples’ Friendship University of Russia, 117198 Moscow, Russia; atyakshin-da@rudn.ru; 2Research Institute of Experimental Biology and Medicine, Burdenko Voronezh State Medical University, 394036 Voronezh, Russia; 3Institute for Hematopathology, Fangdieckstr. 75a, 22547 Hamburg, Germany; verasamoilova@hotmail.com (V.S.); mtiemann@hp-hamburg.de (M.T.)

**Keywords:** CD38, mast cells, tryptase, secretion

## Abstract

The biological significance of the CD38 molecule goes beyond metabolic, enzymatic, and proliferative functions. CD38 possesses the functions of an exoenzyme and receptor, and is actively involved in the mechanisms of adhesion, migration, intercellular signaling, formation of immune synapses, and modulation of the activity of a wide range of immune and non-immune cells. The aim of this study was the immunohistochemical assessment of the cytological and histotopographic characteristics of CD38 expression in mast cells. CD38 expression was found in a minority of the mast cell population. It is characterized by wide variability from low to high levels. The intensity of CD38 expression in mast cells has organ-specific features and depends on the development of pathological processes in a specific tissue microenvironment. The mechanisms of intercellular interaction between mast cells and CD38+ cells foster new understanding of the protumorigenic or antitumor potential of tryptase.

## 1. Introduction

The CD38 molecule was first identified on the surface of T-lymphocytes, and is involved in the mechanisms of cellular activation and proliferation [1,2,3]. Since then, our understanding of the biological significance of CD38 and expression profiles in the cells of a specific tissue microenvironment has significantly expanded [4,5,6]. In addition to functioning as an exoenzyme with a variety of substrates and by-products involved in calcium signaling and metabolic regulation, CD38 is a surface receptor. It is actively involved in the mechanisms of adhesion, migration, and intercellular signaling, including the formation of immune synapses. Receptor properties endow CD38 with the ability to modulate the activity of a wide range of immune and non-immune cells [7,8]. Currently, the cytometabolic functions of CD38—associated with the regulation of mitochondrial activity, increased energy metabolism, its rather pronounced antioxidant properties, and suppression of the mechanisms of inflammation, apoptosis, necrosis, and autophagy—are actively being studied [8]. The role of CD38 goes beyond metabolic, enzymatic, and proliferative functions [4,5,9,10], and it may even be actively involved in morphogenesis and immunogenesis. In this regard, the study of CD38 expression on other cells has become highly relevant. It was found that plasmablasts and plasmacytes are characterized by the highest expressions of CD38, and so this molecule has become a specific molecular marker in immunohistochemical studies [9]. To a lesser extent, CD38 is present in other cells of the lymphoid and myeloid series [10,11,12]. Located on the plasmalemma of T- and B-lymphocytes, CD38 is structurally and functionally associated with other molecules—in particular, TCR, BCR, FcγRIII, CD19, CD16, and related molecules—participating in the initiation of various transcription programs and possessing specific effector functions, including cytotoxic ones, through degranulation and secretion [13,14,15]. Expression of CD38 on cells of the immune system, including B- and T-lymphocytes, dendritic cells, monocytes, macrophages, and granulocytes, can vary depending on the degree of differentiation and the activation status [5]. The expression of CD38 has been studied on non-immune cells, including erythrocytes, platelets, osteoclast and osteoblast precursors, prostate epithelium, kidneys, pancreatic islets of Langerhans, neurons, neuroglia, smooth myocytes, and striated muscles [5].

Despite the data obtained, it is clear that further studies are required into the expression and activity of CD38 in cells of the tumor microenvironment with a pronounced regulatory potential. Among them, mast cells (MCs), which have both protumorigenic and versatile antitumor effects, are of particular importance. MCs have been recognized as key players in the formation and finetuning of antitumor immunity, as well as being modulators of the state of the tumor stroma and extracellular matrix, plus some characteristics of cancer cells. Thus, MCs are a very promising target for cancer immunotherapy [16,17]. However, there is still no literature that addresses histochemical imaging of CD38 in MCs.

Attempts to identify CD38 on mast cells of dispersed tissues of some joyoragnes, carried out at the start of the study of their immunophenotype, were unsuccessful, while the presence of an exoenzyme was shown on basophilic granulocytes [18]. In subsequent flow cytometric studies of normal and pathological bone marrow samples, the expression of CD38 on mast cells was also not detected [19,20]. The only studies investigating the expression of CD38 in MCs were performed using liquid cytometry technology and single-cell RNA-sequencing [21], and showed a high degree of variability of the effect of the exoenzyme on MCs, even within similar tissue localizations. In addition, there are unpublished data on the highly variable effect of CD38 expression on lung mast cells based on analysis using flow cytometry (Preprint article: Rönnberg, E., Boey, D.Z.H., Ravindran, A., Säfholm, J., Orre, A.-C., Al-Ameri, M., Adner, M., Dahlén, S.-E., Dahlin, J.S., and Nilsson, G. Immunoprofiling reveals novel mast cell receptors and a continuous nature of human lung mast cell heterogeneity. bioRxiv [Preprint]. 2021. DOI: 10.1101/2021.03.12.435093). In view of such contradictory data, we decided to assess the effectiveness of immunohistochemical methods for detecting CD38 and studying the intensity of the effect of exoenzyme expression on MCs of various organs, as well as the pathology.

## 2. Materials and Methods

### 2.1. Case Selection

Tissue samples were taken for diagnostic purposes. The biomaterial of the skin of the lower extremities was obtained from male patients diagnosed with neurotrophic ulcers (*n* = 6) or other conditions not accompanied by signs of pathological changes in the skin (*n* = 5). Biopsy material of the mammary glands was obtained to clarify a breast cancer diagnosis; there were six patients with a confirmed diagnosis of cancer and five patients without a confirmed diagnosis. Tonsil tissue was obtained from four male and six female patients undergoing a tonsillectomy for recurrent tonsillitis. The patients were 8–27 years old.

In addition, four patients with indolent systemic mastocytosis were included in the study. The biomaterial was obtained from the archives of the Institute of Hematopathology, Hamburg and the Institute of Pathology, Ludwig-Maximilians University, Munich (Germany).

The samples were redundant clinical specimens that had been anonymized and were not linked to patient information. The study was conducted in accordance with the principles of the Declaration of Helsinki of the World Medical Association (“Ethical Principles for Medical Research with Human Participation”) and was approved by the Supervisory Board of the Institute of Hematopathology, Hamburg, Germany.

### 2.2. Tissue Probe Staining

The biopsy material, after standard fixation in buffered 4% formaldehyde, was embedded in paraffin. The paraffin blocks were cut into 2 µm sections, which were subsequently subjected to standard dewaxing and rehydration procedures, following the standard procedure [22].

### 2.3. Immunohistochemistry

For the immunohistochemical assay, deparaffinized sections were subjected to antigen retrieval by heating the sections in a steamer with an R-UNIVERSAL Epitope Recovery Buffer (Aptum Biologics Ltd., Southampton, UK), at 95 °C for 30 min. According to our earlier recommendations, blocking the endogenous Fc receptors prior to incubation with primary antibodies was omitted [23]. After antigen retrieval, and when required, quenching endogenous peroxidase, sections were immunoreacted with primary antibodies. A list of primary antibodies used in this study is presented in Table 1. For manually performed immunostaining, primary antibodies were applied in concentrations ranging from 1 to 5 µg/mL and incubated overnight at +4 °C. For visualization of the primary anti-CD38 antibodies, we used tyramide signal amplification (TSA) [22].

Bound primary antibodies were visualized using secondary antibodies (purchased from Dianova, Hamburg, Germany, and Molecular Probes, Darmstadt, Germany) conjugated with Cy3 or Alexa Fluor-488. The final concentration of secondary antibodies was between 5 and 10 µg/mL PBS. Both single and multiple immunofluorescence labeling were performed according to standard protocols [22]. A list of the secondary antibodies and other reagents used in this study is presented in Table 2.

To simultaneously detect antigens from the same host species, we performed TSA with a subsequent heat elution treatment after each immunostaining step [24]. The bound primary/secondary antibody complex from the preceding immunolabeling step was eluted with a citrate/acetate-based buffer at pH 6.0, containing 0.3% SDS (also available from VENTANA as CC2 solution, cat #950-223). Nuclei were counterstained with 4′,6-diamidino-2-phenylindole (DAPI, 5 µg/mL in PBS) for 15 s, and the sections were then mounted using VectaShield (Vector Laboratories, Burlingame, CA, USA).

### 2.4. Assessment of CD38 Expression in the Mast Cell Population

To determine the volume of the CD38+ subpopulation of MCs, the presence of the exoenzyme in tryptase-positive MCs was taken into account. Depending on the organ, between 500 and 1000 MCs were examined, from which the relative amount with the expression of CD38 was calculated. From the number of MCs that were identified as CD38-positive, cells with a high, moderate, or weak expression were conditionally identified, and their relative contents were calculated in percentages. The intensity of staining was independently reviewed by three pathologists (DA, IB, and MT) using a ×40 objective lens, and upon reaching consensus, scoring was assigned from (+++) to (+). A consensus score of (+++) represented the highest content of CD38 in the cytoplasm of MC. Weakly noticeable staining of the cytoplasm or cytolemma for CD38 was assigned as (+). The quantification of the cell content was performed using a counting program incorporated in the AxioVision software (Carl Zeiss Vision GmbH, Zeppelinstr. 485399, München-Hallbergmoos, Germany).

## 3. Results

### 3.1. The Effectiveness of Antibodies in the Detection of CD38 in MCs

The effectiveness of detecting CD38 in MCs is dependent on the use of antibodies. Primary rabbit monoclonal antibodies (clone SP149) to the CD38 molecule (Cell Marque, Rocklin, CA, USA) were less effective for detecting CD38 in tryptase-positive MCs. In the tonsils and bone marrow, only a few CD38+ mast cells were detected. In the skin and mammary glands, the number of CD38+ MCs was greater (Table 3). In a pathology, the number of MCs with CD38 expression increased, especially in patients with neurotrophic leg ulcers (Table 3). Occasionally, some intrapopulation MC aggregates with CD38 expression were observed (Figure 1a–a″). At the same time, the intensity of CD38 expression increased, and cells with a very high CD38 content were detected, both on the plasmalemma and in the cytoplasm (Figure 1b,c).

Using antibodies kindly provided by Fabio Malavasi (University of Torino, Italy), the detection efficiency of CD38+ MCs was higher in all organs. CD38+ MCs were more frequently detected in the mammary glands (Figure 1d) and were also found in the bone marrow and tonsils (Figure 1e,e′). At the same time, the highest frequency of immunohistochemical detection of CD38 was characterized by the population of MCs in the skin (Figure 1f,f′,g,h,i,i′). In mastocytosis, MCs differed in their levels of CD38 content. The pronounced expression in some of the CD38 levels attracted attention. Hypochromic type-1a MCs, which have an elongated shape, showed the highest expression of CD38, especially at the plasma membrane (Figure 2a). The greatest number of CD38+ MCs was detected in the skin of neurotrophic ulcers, both in small and large MCs filled with large tryptase-positive granules (Figure 2b). Although the lowest number of CD38+ MCs was detected in the tonsilla, their more frequent co-localization with other CD38+ cells of the lymphoid tissue was found (Figure 2c, Figure 3a and Figure 4). In some cases, the process of penetration of tryptase-positive granules into the cytoplasm of CD38+ cells was visualized (Figure 4a,a′,b,c,d).

Using antibodies kindly provided by Fabio Malavasi, University of Torino, Italy, CD38+ MCs were more frequently detected in the mammary glands (Figure 1d), and were also found in the bone marrow and tonsils (Figure 1e,e′). At the same time, the highest frequency of immunohistochemical detection of CD38 mirrors the high population of MCs in the skin (Figure 1f,f′,g,h,i,i′).

In mastocytosis, MCs differed in their CD38 levels. The pronounced expression of CD38 in some of them attracted attention. Hypochromic type-1a MCs, which have an elongated shape, showed the highest expression of CD38, especially on the plasma membrane (Figure 2a). The greatest number of CD38+ MCs was detected in the skin of neurotrophic ulcers, both in small and large MCs filled with large tryptase-positive granules (Figure 2b). Despite the fact that the lowest number of CD38+ MCs was detected in the tonsils, their more frequent co-localization with other CD38+ cells of the lymphoid tissue was found (Figure 2c, Figure 3a and Figure 4). In some cases, the process of penetration of tryptase-positive granules into the cytoplasm of CD38+ cells was visualized (Figure 4a,a′,b,c,d).

Attention should be paid to the clear relationship between CD38 expression and the state of the specific tissue microenvironment. In particular, during inflammation or tumorigenesis, the expression of CD38 in mast cells is significantly increased (Figure 2d,d′,e,e′). In patients with neurotrophic ulcers, MCs had the highest expression of CD38 (Figure 2f,f′,g,g′). In the mammary gland, along with an increase in the expression of CD38 in mast cells during carcinogenesis (Table 1), there was a significant difference in the preferential localization of the exoenzyme (Figure 2d,d′,e,e′,f,f′,g,g′).

### 3.2. CD38 Cytotopography in Mast Cells

Variants of cytotopography of the predominant expression of CD38 in MCs should be noted. In some of them, CD38 was exclusively identified on the plasmalemma (Figure 1d,f,f′,g,h,i,i′ and Figure 2b,d,d′,e,e′); in others, the expression was primarily characterized by an intracytoplasmic distribution (Figure 1c–c″,e,e′ and Figure 2c–c″,f,f′,g,g′). Finally, there were cells where the expression of CD38 was detected both on the plasma membrane and in the structures of the cytoplasm (Figure 1b and Figure 2a).

From the point of view of cytotopographic localization, primary antibodies provided by Fabio Malavasi (University of Torino, Italy) detected CD38 mainly in the area of the plasma membrane, although in some cases, intracytoplasmic localization of this exoenzyme was also noted. The rabbit monoclonal antibody of another manufacturer, Cell Marque, along with the detection of the exoenzyme on the plasma membrane, detected CD38 in cytoplasmic structures with a higher frequency.

In some cases, it was possible to draw an analogy between the granules and exoenzyme content. At the same time, diffuse staining of the cytoplasm was observed much more frequently, without any precise intracytoplasmic localization of the enzyme (Figure 2c,f,f′,g,g′).

Attention was drawn to the regularity of the rarer detection of CD38 in MCs containing well-defined granules. In mastocytosis, CD38 is more often expressed in hypochromic MCs containing small granules [25].

It is possible that the intracytoplasmic distribution of CD38 also arises in MCs that have a pronounced expression of the exoenzyme on the plasmalemma; however, in terms of dynamic characteristics, the camera makes it difficult to register a weaker signal in the background of a more pronounced fluorescent signal from other cells in the same field of view. Thus, even with a pronounced pattern of CD38 distribution on the plasmalemma, it can be assumed that there is a small amount of CD38 in the cytoplasm.

### 3.3. Histotopographic Features of Co-Localization of Mast Cells and CD38+ Cells in a Specific Tissue Microenvironment

Of great interest from the point of view of the implementation of intercellular signaling is the co-localization of MCs with and without expression of CD38 (Figure 2a). The possible close setting of such MCs suggests previously unknown mechanisms of intrapopulation interaction, and thus consideration of this exoenzyme as a direct participant in the directed induction in migration activity, sensitivity of reception, and secretory activity.

Co-localization of MCs with other CD38+ cells of the specific tissue microenvironment of organs is of particular interest (Figure 3d–k and Figure 4). This necessitates objective study of the spatial co-localization and analysis of intercellular interactions, taking into account the components of the secretome of MCs. The analysis of such data reveals morphological evidence of the implementation of signaling pathways and induction of the formation of immunocompetent cell ensembles according to the direction of the formation of a protumor or antitumorigenic microenvironment. Sometimes, MCs coincidentally made contact with several CD38+ cells (Figure 3f,h and Figure 4a,g). This can be considered from the point of view of implementing selective degranulation of MCs with the formation of a specific immunological synapse at a certain cell locus, characterized by a certain molecular clustering, segregation, and directed secretion of the necessary components of the secretome [26]. In this context, the local effect of specific mast cell proteases, including tryptase, during intercellular contact is of particular informational value. Such polarized degranulation within an immunological synapse with targeted tryptase secretion can be identified on microscope slides (Figure 4).

### 3.4. Content of Tryptase in Nuclei of CD38+ Cells

Normally, finding mast cell tryptase in the nuclei of CD38+ cells is quite rare. Most often, this was detected in the tonsils (Figure 3a). At the same time, in a pathology, the frequency of tryptase detection in the nuclei of other cells increased, particularly in patients with neurotrophic ulcers or in oncogenesis (Figure 3b–d). Sometimes, MCs had direct contact with CD38+ cells where the nuclei contained tryptase (Figure 3a). Interestingly, within CD38+ cells, cells with tryptase-positive nuclei were sometimes found without any presence of MCs nearby (Figure 3b,c). This suggests the supply of tryptase from extracellular protease resources, the source of which could be mast cell granules freely located in the connective tissue. Mast cell granules could be located in the extracellular matrix for a certain time, possess a certain autonomy, and represent a reserve of a specific protease.

From the point of view of explaining the mechanism of tryptase entry into the nuclei of CD38+ cells, we can assume the regulatory consequences of this phenomenon. In particular, this phenomenon can affect the state of histones and cell proliferation, while tryptase exhibits antiproliferative effects [27,28].

## 4. Discussion

Since the discovery of CD38 almost four decades ago, accumulated data indicate its important role in the activity of various cell types in both physiological and pathological contexts [8]. Human multipotent blood stem cells can proliferate and differentiate into mature MCs [29], while the committed MC precursors contain the CD38 antigen [30,31]. CD38 is expressed at an early stage of CD34+ stem cell differentiation and is preserved in mature immune cells, including T cells and NK cells, B cells and plasma cells, granulocytes, etc. Expression of CD38+ in neutrophils provides important confirmation of the possibility of identifying this molecule in MCs because neutrophils and MCs share a common bone marrow precursor. The possibility of detecting CD38 in mature MCs using flow cytometry has been shown in several studies, despite the high variability of exoenzyme expression [21]. At the same time, according to the results of scRNA-seq, various phenotypes of airway MCs with high and low expression levels of CD38 (CD38^high^CD117^low^ and CD38^low^CD117^high^) were revealed. The authors suggested that these immunophenotypes are two polarized states with specific transcriptome, proteome, and histotopographic characteristics [21]. In addition, the authors suggested that MCs with the CD38^high^CD117^high^ immunophenotype, which were detected in small numbers, represent an intermediate phenotype [21].

The results of our immunohistochemical analysis confirmed the possibility of identifying CD38 in human MCs. At the same time, the question of the specificity of the antibodies used remains fundamental since the different efficacies of antibodies clearly arises due to their affinity for various epitopes of the CD38 molecule. The importance of this fact in the context of multiplex immunohistochemistry is evidenced by the results of our previous work, which showed the unequal efficiency of CTLA-4 detection using various antibodies [32]. Our results showed both differences in detecting CD38 in MCs, depending on the antibodies used, and different intensities of exoenzyme expression. The data obtained are consistent with the results of a study that revealed the effects of targeted therapy in accordance with the unequal affinity of antibodies to various epitopes of CD38, including the mechanisms of induction of apoptosis of target cells [33,34].

An interesting point for discussion may be the regularity of the co-expression of CD117 and CD38 in MCs. Indeed, judging by our data, it is possible to reveal a certain analogy between the numbers of CD117+ MCs and CD38+ MCs in an organ-specific population. Although CD117 is a reliable marker of MCs, its detection does not guarantee the determination of the total population of MCs in an organ. There may be a definite transition between these two mast cell immunophenotypes. Further studies of the polarization of MCs in relation to the expression of the exoenzyme CD38 and the protein tyrosine kinase KIT (CD117) may yield interesting results for the interpretation of the functional state.

To assess the cytotopographic distribution of CD38, it is necessary to consider that the expression of this exoenzyme depends on the ultrastructural configuration of the molecule and its intracellular topography. In addition to the plasmalemma, CD38 was also identified in exosomes, in which it retained its enzymatic activity and extracellular microvesicles, which also contained other components of the adenosinergic apparatus, including CD203a, CD39, and CD73 [8]. This fact creates the prerequisites for studying CD38 in exosomes and mast cell microvesicles, to understand and interpret intercellular communication. The intracellular pool of CD38 is associated with mitochondrial and nuclear membranes [35]; this may be because several structures in the cytoplasm of MCs are immunopositive for the exoenzyme (Figure 2f,f′,g,g′). At the same time, a certain intracellular level of CD38 can be formed due to the mechanism of internalization of CD38, leading to the distribution of the exoenzyme in the cytoplasm.

The intracellular distribution of CD38 can be determined in relation to the level of phagocytotic activity. In particular, CD38 is found in the emerging phagosomes of type-1 macrophages [36]. Since MCs have the functionality for phagocytosis, a high expression of CD38 in some of their pool may have additional functional properties in relation to the development of relationships with tumors or other pathological agents. Thus, the intracellular distribution of CD38 may be an additional marker for assessing the biological effects of MCs in the tumor microenvironment.

In an immunohistochemical study, the intense expression of CD38 on the plasmalemma provides a fairly reliable marker for the identification of plasma cells [11]. At the same time, activated B cells and T cells, including NK cells, can increase the expression of CD38 to the level of plasma cells [37]. This pattern was also revealed in the study of monocytes, macrophages, dendritic cells, and granulocytes. It is clear that in MCs, the expression level of CD38 is also a labile trait and can vary depending on the state of the specific tissue microenvironment.

To evaluate the capabilities of multiplex immunohistochemistry for assessing the level of CD38 expression, it is necessary to consider that a low visualized level of the exoenzyme in MCs may be associated with dynamic characteristics of the monochrome camera operation when recording immunohistochemical staining using fluorochromes. The camera adapts to a signal with a high level of luminescence, and if the level of expression in neighboring targets is low enough, then they can remain poorly visible or even be invisible.

Studies of the role of CD38 in the progression of oncogenesis and mechanisms of tumor evasion from immunity are ongoing. Several studies have shown the importance of CD38 as a factor contributing to tumor development due to a change in the local metabolic microenvironment of the tumor from pro-inflammatory to anti-inflammatory, including the polarization of macrophages [38,39]. Understanding CD38 in this way allows us to consider CD38 as a marker for monitoring the antitumor immune response, and thus determining the metabolic profiles of tissues [36]. The different expressions of CD38 in MCs may reflect their physiological states or degrees of participation in oncogenesis, or indicate the targeted secretion of cytokines, chemokines, and growth factors with pro-oncogenic or antitumorigenic effects.

CD38 can also determine the activity of mast cell migration; its participation in the efficiency of chemotaxis of other human immunocompetent cells is well-known [5].

MCs have a rich arsenal for the regulation of other immune cells both through direct contact and by acting indirectly [40]. Our results indicate an important diagnostic significance of events associated with co-localization of MCs and CD38+ cells. Of considerable interest is the formation of intercellular contact, particularly in a specialized mast cell signaling region called the ADDS (antibody-dependent degranulatory synapse) in connection with local degranulation [26]. The secretion by MCs in this area can have negative consequences for the co-localized cells, in particular, causing lysis by TNF, caspase 3, or granzyme B [26]. MCs can cause cytotoxic damage to other cells using exosomes, secretory granules, and macro- and microvesicles. In this case, contact cells can capture exosomes and induce their cytolysis with the participation of specific mast cell proteases [26]. In addition to this effect, we should consider the inductive functions of tryptase exerted on the proliferative potencies of the immune contact cell [27,28]. Moreover, the activating effects on the secretion of granzymes and cytokines in NK cells with the acquisition of a cytotoxic phenotype are well-known [41]. Thus, the identified immunological synapses between CD38+ cells and MCs may reflect complex intercellular signaling mechanisms and could present an indirect marker of the effectiveness of antitumor therapy.

## 5. Conclusions

Our study shows the principal possibility of detecting CD38 in MCs in histological preparations. In this case, antibodies directed to certain epitopes of CD38 are important, which is associated with their specificity and effectiveness in detecting MCs. The intensity of CD38 expression in MCs is related to their functional activity. The details and features of this mechanism could provide fertile ground for future research in various laboratories. On the other hand, CD38+ MCs can represent separate subpopulations with their own special properties, which also deserve separate studies using omics technologies, including genomic, transcriptome, proteomic, lipidomic, etc. The more information obtained regarding the multifaceted protein CD38 and MCs in the tumor microenvironment, the more focused the suppression of CD38 activity can be in regard to the efficiency of tumor eradication.

## Figures and Tables

**Figure 1 cells-10-02511-f001:**
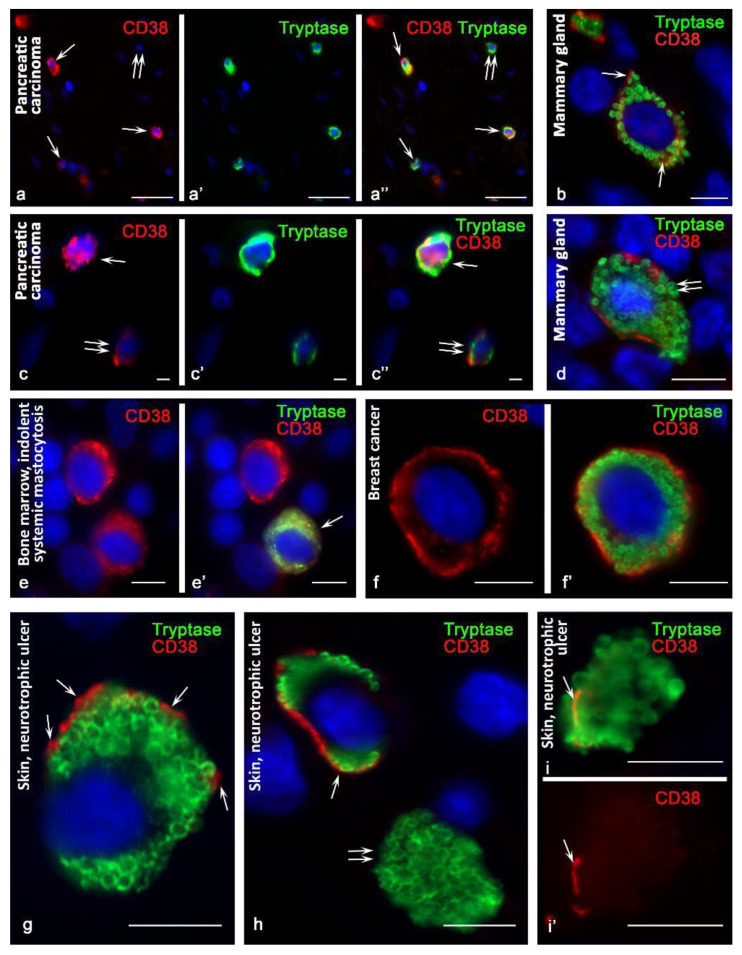
CD38 cytotopography in mast cells. Primary antibodies used: (**a**–**e**) rabbit monoclonal (Cell Marque, USA); (**f**–**i**) mouse monoclonal (kindly provided by Fabio Malavasi, University of Torino, Italy). (**a**–**a″**) Pancreatic carcinoma. Cells with expression (arrowed) and without expression (double arrow) of CD38 are identified. (**b**) The mammary gland. Expression of CD38 in the cytoplasm and on the plasma membrane (indicated by an arrow). (**c**–**c″**) Pancreatic carcinoma. MCs with high (arrow) and moderate (double arrow) CD38 expression. (**d**) The mammary gland. The predominantly peripheral location of CD38 in the area of the plasma membrane, with varying severity along its length. At the site of secretion of tryptase, CD38 is absent (double arrow). (**e**,**e′**) Indolent systemic mastocytosis, bone marrow. Diffuse distribution of CD38 in the cytoplasm of a hypogranulated mast cell (indicated by an arrow). (**f**,**f′**) Breast cancer. Preferential peripheral location of CD38. (**g**) Skin. Peripheral localization of CD38 at some loci of the plasma membrane (indicated by an arrow). (**h**,**i**,**i′**) Skin, neurotrophic ulcer. (**h**) MCs with CD38 expression on plasmalemma (arrowed) and absent (double arrow). (**i**,**i′**) Expression of the exoenzyme at the peripheral locus of the non-nuclear cytoplasmic fragment of the mast cell (indicated by the arrow). Scale bar: 50 µm (**a**–**a″**), 5 µm (all others).

**Figure 2 cells-10-02511-f002:**
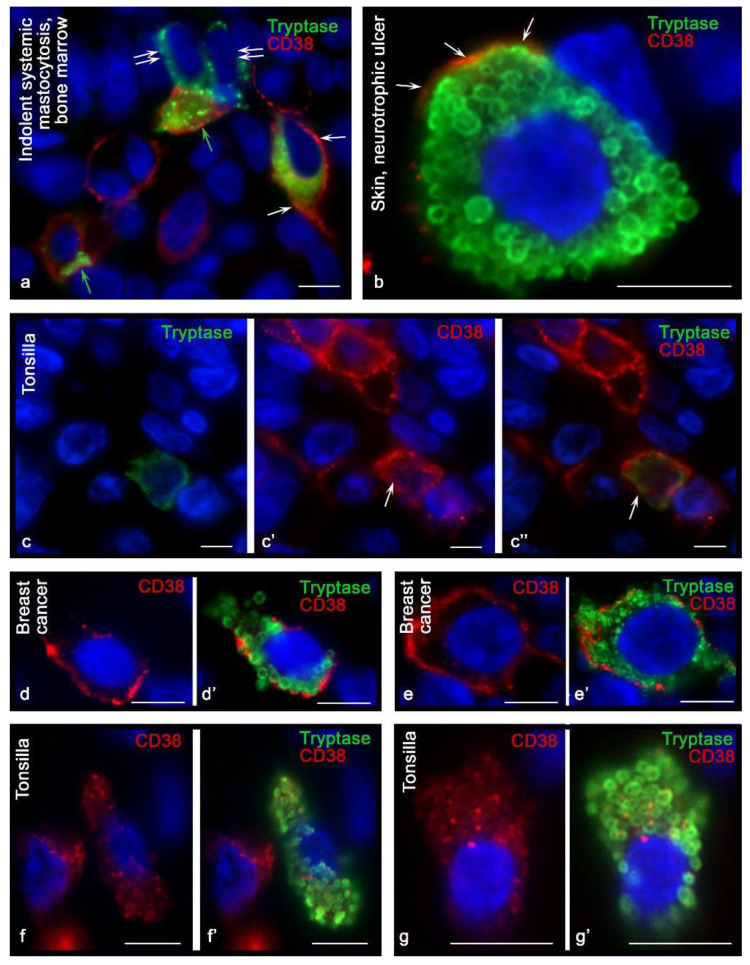
Intracellular distribution of CD38 in mast cells. Primary antibodies used: (**a**,**b**,**d**,**e**) mouse monoclonal (kindly provided by Fabio Malavasi, University of Torino, Italy); (**c**,**f**,**g**) rabbit monoclonal (Cell Marque, USA). (**a**) Indolent systemic mastocytosis, bone marrow. In the type-1a hypochromic mast cell, the predominant localization of CD38 on the plasma membrane is revealed (indicated by the arrow). MCs with CD38 localization on intracellular structures (green arrow) and lack of exoenzyme expression (double arrow) are found. (**b**) Skin, neurotrophic ulcer. Expression of CD38 on the mast cell plasmalemma is unequal, with the highest expression at some loci (indicated by an arrow). (**c**–**c″**) Extrafollicular lymphoid tissue of the tonsil. In the mast cell, predominantly intracytoplasmic expression of CD38 is detected (indicated by the arrow). (**d**,**d′**,**e**,**e′**) Breast cancer. Variants of CD38 distribution on mast cell plasmalemma. (**f**,**f′**,**g**,**g′**) Tonsilla. Stroma. Predominantly intracytoplasmic expression of CD38. Scale bar: 5 µm.

**Figure 3 cells-10-02511-f003:**
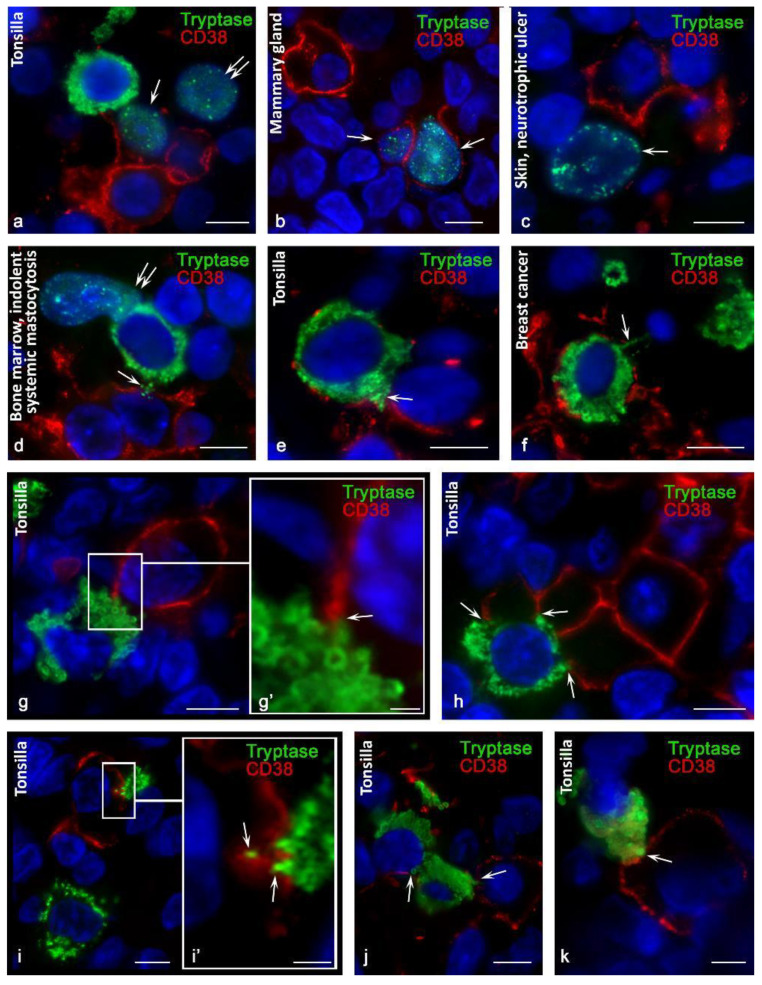
Features of the interaction of mast cells with CD38+ cells in a specific tissue microenvironment. Primary antibodies used: (**a**,**b**,**h**–**k**) rabbit monoclonal (Cell Marque, USA); (**c**–**g**) mouse monoclonal (kindly provided by Fabio Malavasi, University of Torino, Italy). (**a**) Tonsilla. Mast cell in contact with CD38+ cells; localization of tryptase in the nucleus of CD38-positive and CD38-negative cells (arrow and double arrow, respectively). (**b**) The mammary gland. The location of mast cell tryptase in the nuclei of CD38+ cells (indicated by an arrow). (**c**) Skin, neurotrophic ulcer. Tryptase in the nucleus of CD38-negative cell. (**d**) Indolent systemic mastocytosis, bone marrow. Directed secretion of mast cell tryptase into a CD38-positive cell (arrow); tryptase in the nucleus of an adjacent CD38-negative cell (double arrow). (**e**) Tonsilla. The CD38+ mast cell makes contact over a large area with the CD38+ cell (indicated by the arrow). (**f**) Breast cancer. Mast cell in close contact with CD38+ cells; secretion of tryptase towards the nucleus of the adjacent cell (arrow). (**g**–**k**) Tonsilla. Various variants of tryptase secretion and contact of mast cell granules with the plasmalemma of CD38+ cells (indicated by an arrow). Scale bar: 1 µm (**g′**,**i′**), 5 µm (all others).

**Figure 4 cells-10-02511-f004:**
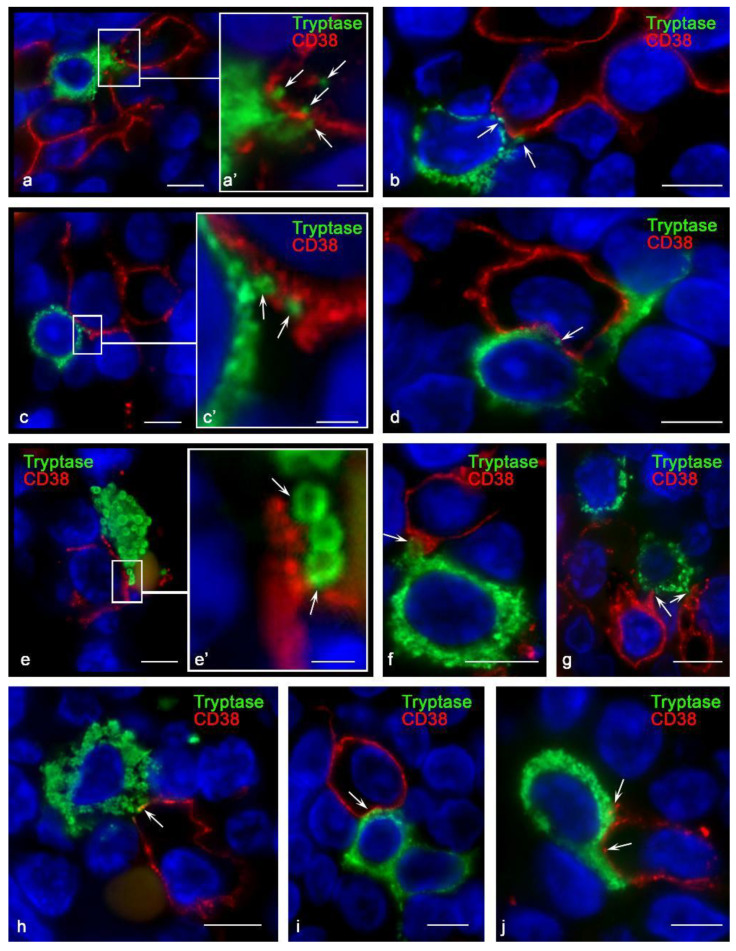
Histoarchitectonics of specialized immune contact of mast cells with CD38+ cells in tonsilla. Primary antibodies used: (**a**–**e**) rabbit monoclonal (Cell Marque, USA); (**f**–**j**) mouse monoclonal (kindly provided by Fabio Malavasi, University of Torino, Italy). (**a**,**a′**) Mast cell co-localization with multiple CD38+ cells. The enlarged fragment (**a′**) shows the entry of tryptase-positive granules into the cytoplasm of CD38+ cells towards the nucleus (indicated by the arrow). (**b**,**c**,**c′**,**d**) Contact of a mast cell with CD38+ cells; various variants of co-localization of tryptase-containing granules with the plasmalemma of CD38+ cells (indicated by an arrow). (**e**) Interaction of tryptase-positive granules with an exoenzyme on the plasma membrane of a co-localized cell (indicated by an arrow). (**f**–**j**) Different morphological variants of the formation of the interaction of mast cell granules with CD38+ tonsillar cells (indicated by an arrow). Scale bar: 1 µm (**a′**,**c′**,**e′**), 5 µm (all others).

**Table 1 cells-10-02511-t001:** Primary antibodies used in this study.

Antibody/Clone	Host	Source	Dilution
CD38 (SP149)	Rabbit monoclonal	Cell Marque, Rocklin, USA	1:300
CD38 (IB4; IB6, SUN-4B7)	Mouse monoclonal	Antibody kindly made available by Fabio Malavasi, University of Torino, Torino, Italy	1:200
Anti-mast cell tryptase antibody [AA1] #ab2378	Mouse monoclonal	Abcam, UK	1:3000
Anti-mast cell tryptase antibody [EPR9522] #ab151757	Rabbit monoclonal	Abcam, UK	1:2000

**Table 2 cells-10-02511-t002:** Secondary antibodies and other reagents.

Antibodies and Other Reagents	Source	Dilution	Label
Goat anti-mouse IgG Ab (#115-165-166)	Jackson ImmunoResearch, Ely, United Kingdom	1/200	Cy3
Goat anti-rabbit IgG Ab (#A-11034)	Invitrogen, Darmstadt, Germany	1/200	Cy3
Goat anti-mouse IgG Ab (#A-11029)	Invitrogen, Darmstadt, Germany	1/200	Alexa Fluor 488
Goat anti-rabbit IgG Ab (#A-11034)	Invitrogen, Darmstadt, Germany	1/200	Alexa Fluor 488
CC2 solution (#950-223)	Ventana Medical Systems, Lund Sweden	Ready-to-use	w/o
VECTASHIELD mounting medium (#H-1000)	Vector Laboratories, Burlingame, CA, USA	Ready-to-use	w/o
TSA plus fluorescein (#NEL741E001K)	PerkinElmer, Rodgau, Germany	1/200	Fluorescein
TSA plus cyanine 3 (#NEL744E001KT)	PerkinElmer, Rodgau, Germany	1/200	Cyanine 3

**Table 3 cells-10-02511-t003:** Efficiency of immunomorphological detection of CD38 in MCs.

Organ	Primary Antibodies to CD38 (SP149) Rabbit Monoclonal Antibody (Cell Marque, USA)				Primary Murine Antibodies to CD38 (Kindly Provided by Fabio Malavasi, University of Torino, Italy)			
	Number CD38+ MC (%, *M ± m*)	Expression Intensity (%)			Number CD38+ MC (%, *M ± m*)	Expression Intensity (%)		
		+ (*M* ± *m*)	++ (*M* ± *m*)	+++ (*M* ± *m*)		+ (*M* ± *m*)	++ (*M* ± *m*)	+++ (*M* ± *m*)
Skin (normal; *n* = 5)	5.2 ± 0.5	65.2 ± 3.1	22.3 ± 1.9	12.5 ± 1.1	8.8 ± 0.5	52.4 ± 4.2	39.2 ± 2.5	8.4 ± 0.5
Skin (neurotrophic ulcers; *n* = 6)	7.3 ± 0.6 ^∆^	13.2 ± 0.8 ^∆^	62 ± 4.3 ^∆^	24.8 ± 1.6 ^∆^	14.1 ± 1.2 ^∆^	15.6 ± 1.1 ^∆^	66.7 ± 4.3 ^∆^	17.7 ± 1.3 ^∆^
Breast (normal; *n* = 5)	3.2 ± 0.2	58.9 ± 5.1	32.7 ± 2.8	8.4 ± 0.9	8.4 ± 0.6	45.6 ± 3.4	47.1 ± 3.1	7.3 ± 0.5
Mammary cancer (*n* = 6)	3.1 ± 0.3	23.5 ± 1.9 ^∆^	61.3 ± 5.3 ^∆^	15.2 ± 1.4 ^∆^	12.3 ± 1.3 ^∆^	35.6 ± 2.2 ^∆^	55.6 ± 3.2 ^∆^	8.8 ± 0.4
Bone marrow (mastocytosis; *n* = 4)	*	-	-	-	8.2 ± 0.6	65.2 ± 4.3	29.5 ± 1.8	5.3 ± 0.3
Tonsilla (recurrent tonsillitis; *n* = 10)	*	-	-	-	9.7 ± 0.5	59.4 ± 4.1	35.9 ± 2.2	4.7 ± 0.4

Legend: *, single cells; +, weak expression; ++, moderate expression; +++, pronounced expression; ^∆^, *p* < 0.05.

## Data Availability

All data and materials are available upon reasonable request. Address requests to I.B. (email: buchwalow@pathologie-hh.de) or M.T. (email: mtiemann@hp-hamburg.de) of the Institute for Hematopathology, Hamburg, Germany.

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
