# Peer review of "Expression of CD38 in Mast Cells: Cytological and Histotopographic Features"

_cells, 2021, doi:10.3390/cells10102511_

Round 1

Reviewer 1 Report

In their study Atiakshin and colleagues have investigated the expression of CD38 in human mast cells by classical immunohistology. They compared mast cells of different tissues and diseases and analyzed histotopographic features of CD38 along with tryptase as a mast cell-specific enzyme in a semiquantitative manner. This study is of interest in the field leukemia and chronic inflammatory diseases, where CD38 is a novel therapeutical target.

Authors referred to the fact that mast cells are able to shape the tumor microenvironment and therefore, analysis of CD38 is worth to be analyzed.

Indeed, CD38 is expressed in a plethora of immune cells and non-immune cells, but its expression in mast cells has been only investigated in an earlier studies of Valet et al., 1989. Surprisingly, in this flow cytometric study of dispersed tissue samples CD38 was found to be expressed in basophils but not in mast cells. This study should be discussed/included in their manuscript, at least in the introduction where only two actual references have been cited.  Only the study of Dwyer et al., 2021 once again analyzed the expression of CD38 in human mast cells. They identified a small subset of subepithelial mast cells co-expressing tryptase, chymase and CD117 in allergic patients. I was wondering that Atiakshin and colleagues did not include at least chymase and CD117 in their staining panel to discuss parallel findings in their tissue samples.

Unfortunately, the general concept of this study is poor because the most interesting question which subtype of mast cells do express CD38 independently from the tissue investigated can not be addressed by the experimental design of the study. Unfortunately, no other immunophenotypic markers besides tryptase has been analyzed. So, it can be only speculated if CD38 is an indication for activated mast cells. At least CD63 or CD107a should have been included in their immunhistological staining panel.

In the first part of the results authors compared two different CD38-specific monoclonals with respect to their staining capacity and identified the murine clone as the more efficient one. But here additional information about antibody titrations of primary and secondary antibodies has to be considered; since differences detected and summarized in table 3 are not striking and no statistical group comparisons have been performed. I would suggest to provide these data rather as supplementary material. According to the statistics summarized in table 3 it is unclear whether the mean number of CD38+ cells is related to the absolute number auf counted tryptase-positive cells of a single individual or the mean number of all individuals analyzed, i.e. six patients with neurotrophic ulcers and 5 control patients, respectively? I would expect mean numbers for every individual analyzed.

How many fields of view were analyzed per sample; how many mast cells were detected at all per field of view and finally please indicate these numbers for all tissue samples investigated (Supplementary table). It would be interesting to know the inter-individual variation within “normal” skin samples and diseased samples.

Due to the limited number of patients included it will be doubtful that-disease-specific staining patterns of CD38-expressing mast cells can be shown. Therefore, authors should focus on the description of tissue-specific mast cells expressing CD38 rather than on possible disease-specific patterns.

In summary, this descriptive study shows that there is a small subset of mast cells expressing CD38, but questions tackling origin, phenotype, function, turnover and their pathogenetic role remain unaddressed.

Author Response

To Reviewer 1

The authors' response to Comments and Suggestions for Authors is given below and highlighted italic:

In their study Atiakshin and colleagues have investigated the expression of CD38 in human mast cells by classical immunohistology. They compared mast cells of different tissues and diseases and analyzed histotopographic features of CD38 along with tryptase as a mast cell-specific enzyme in a semiquantitative manner. This study is of interest in the field leukemia and chronic inflammatory diseases, where CD38 is a novel therapeutical target.

The authors are grateful to the Reviewer for assessing the purpose of the study. Indeed, we first attempted to analyze the expression of CD38 in mast cells on histological preparations, that is, on biomaterial that was subjected to classical histoprocessing procedures with embedding in paraffin. Previously, no one has carried out such work.

Authors referred to the fact that mast cells are able to shape the tumor microenvironment and therefore, analysis of CD38 is worth to be analyzed. Indeed, CD38 is expressed in a plethora of immune cells and non-immune cells, but its expression in mast cells has been only investigated in an earlier studies of Valet et al., 1989. Surprisingly, in this flow cytometric study of dispersed tissue samples CD38 was found to be expressed in basophils but not in mast cells. This study should be discussed/included in their manuscript, at least in the introduction where only two actual references have been cited.  Only the study of Dwyer et al., 2021 once again analyzed the expression of CD38 in human mast cells. They identified a small subset of subepithelial mast cells co-expressing tryptase, chymase and CD117 in allergic patients.

The authors are grateful to the Reviewer for detailed information on previous studies on the identification of CD38 on mast cells. CD38 negative in the study by Valent et al. (1989) may be related to the primary antibodies used OKT10 (CD38), purchased from Ortho Pharmaceuticals, Raritan, NJ. Unfortunately, we were unable to test the effectiveness of these old outdated antibodies with complicated availability. In addition, the authors (Valent et al., 1989) obtained mast cells from dispersed organ tissues, whereas we did not destructed the anatomical integrity of tissues under study in our experiment on paraffin sections in situ. On the recommendation of the Reviewer, we have included in the bibliography the article “Valent P, Ashman LK, Hinterberger W, Eckersberger F, Majdic O, Lechner K, Bettelheim P. Mast cell typing: demonstration of a distinct hematopoietic cell type and evidence for immunophenotypic relationship to mononuclear phagocytes. Blood. 1989 May 15; 73 (7): 1778-85 ", as well as Escribano L, et al., 1998 on the study of bone marrow mast cells.

I was wondering that Atiakshin and colleagues did not include at least chymase and CD117 in their staining panel to discuss parallel findings in their tissue samples.

The aim of our work was not to repeat the study conducted by Dwyer et al. (2021), Our study was aimed on the fundamental possibility of detecting the expression of CD38 on mast cells of histological sections using an immunohistochemical method of research, with the identification of organ-specific characteristics. At this point, the authors thank the Reviewer for his idea to analyze the features of CD38 expression in mast cells of the respiratory mucosa depending on the expression of chymase or CD117. This might be the basis of a separate independent study that will allow us to compare our results with those of the study by Dwyer et al. (2021).

Unfortunately, the general concept of this study is poor because the most interesting question which subtype of mast cells do express CD38 independently from the tissue investigated can not be addressed by the experimental design of the study. Unfortunately, no other immunophenotypic markers besides tryptase has been analyzed. So, it can be only speculated if CD38 is an indication for activated mast cells. At least CD63 or CD107a should have been included in their immunhistological staining panel.

The authors thank the Reviewer for the remark made, but want to emphasize once again that the main task of our study was the fundamental possibility of detecting CD38 on mast cells using immunohistochemical staining technology. Previously, in numerous studies, this exoenzyme was identified on many myelopoiesis cells, which made it possible to create a number of concepts and hypotheses of the physiology and pathology of CD38 in tissues. However, so far in the literature, we have found only two works in which the expression of CD38 on mast cells of the respiratory organs has been shown (Article preprint: Rönnberg, E. et al., 2021, and also Dwyer, DF et al., 2021), and there is not a single work that analyzes the effectiveness of immunohistochemical detection. But we carried out this work and made sure that tryptase-positive mast cells with high expression of CD38 exist in the mast cell population of some organs.

The use of tryptase in our work is due to the fact that the detection of this protease is the most objective method for identifying the total volume of mast cells in an organ. This is evidenced by both the publications of different authors and the results of their own research (Atiakshin et al., 2017). That is why we have chosen tryptase as a mast cell marker in the study of CD38 expression.

In the following, we will gratefully use the advice of the reviewer to expand the study of the cytomorphological features of mast cells expressing CD38. In particular, we are going to investigate the co-localization of mast cell activation markers and CD38 for a more detailed assessment of the characteristics of organ-specific mast cell populations. The study of the expression of mast cell activation markers CD63 and CD107a with the expression of CD38 will significantly expand the analysis of formalin-fixed and paraffin-embedded sections to assess the immunophenotype of mast cells.

In the first part of the results authors compared two different CD38-specific monoclonals with respect to their staining capacity and identified the murine clone as the more efficient one. But here additional information about antibody titrations of primary and secondary antibodies has to be considered; since differences detected and summarized in table 3 are not striking and no statistical group comparisons have been performed.

The authors thank the Reviewer for the comment and note that Tables 1 and 2 indicate the dilutions of the antibodies that were used for the detection of CD38.

I would suggest to provide these data rather as supplementary material. According to the statistics summarized in table 3 it is unclear whether the mean number of CD38+ cells is related to the absolute number auf counted tryptase-positive cells of a single individual or the mean number of all individuals analyzed, i.e. six patients with neurotrophic ulcers and 5 control patients, respectively?

The authors are grateful to the Reviewer for the comment regarding the insufficiently clear presentation of the data in Table 3. Your comments were taken into account in the table, in which we indicated the number of patients in parentheses, and also named the columns M ± m. We actually calculated M ± m in each patient group. We did not set the task of comparing the significance of differences in the effectiveness of using antibodies from different manufacturers, but we supplemented Table 3 with data on the significance of differences in the expression of CD38 (skin and mammary gland) in health and during the development of oncogenesis.

I would expect mean numbers for every individual analyzed.

The authors thank the Reviewer for this comment. However, in our opinion, the introduction of these data into the publication would unnecessarily overload the text without a noticeable increase in the information content of the results presented.

How many fields of view were analyzed per sample; how many mast cells were detected at all per field of view and finally please indicate these numbers for all tissue samples investigated (Supplementary table). It would be interesting to know the inter-individual variation within “normal” skin samples and diseased samples.

The authors are grateful to the Reviewer for specifying the results of the study. In our work, we did not focus on the number of visual fields, since their composition could include structures in which there could not be mast cells (for example, the lumen of a vessel, or a hair follicle in the skin, etc.). However, it should be noted that in the “Materials and Methods” section, we indicated the range of the number of cells that we used to obtain the relative data on the expression of the exoenzyme: from 500 to 1000 for the organ tissue of each patient. This was enough to understand the relative abundance of mast cells with CD38 expression in the general organ-specific population.

As we already mentioned, on the advice of the reviewer, we added to the table the significance of differences in CD38 expression between norm and pathology. In our opinion, this is an important criterion for changing the expression of this exoenzyme under pathological conditions.

Due to the limited number of patients included it will be doubtful that-disease-specific staining patterns of CD38-expressing mast cells can be shown. Therefore, authors should focus on the description of tissue-specific mast cells expressing CD38 rather than on possible disease-specific patterns.

The authors agree with the reviewer for the remark made. However, one of the main goals of our work was not an accurate analysis of CD38 expression in a specific pathology, but to show the possible range of changes in this indicator in pathology (oncogenesis).

In summary, this descriptive study shows that there is a small subset of mast cells expressing CD38, but questions tackling origin, phenotype, function, turnover and their pathogenetic role remain unaddressed.

The authors agree with the reviewer that there exist a wide field for future independent studies of the biology of CD38 mast cells.

The authors thank the Reviewer for useful comments.

Igor Buchwalow

Reviewer 2 Report

The authors present an immunohistochemical evaluation of the cytological and histotopographic characteristics of CD38 expression in mast cells. The authors found that CD38 expression in a minority of the mast cell population. They indicate that CD38 expression is characterized by wide variability from low to high levels.

Results, in Table 3 must include the n corresponding to the number of assessment in each organ.

Results, it is necessary that the figures maintain consistency with the symbols. The arrow must indicate the same type of result, and the double arrow a different result. The symbols should indicate the same type of observation in each of the figures.

Moreover, for a better understanding of the results obtained, it would be necessary for the authors to indicate in each figure to which organ each image corresponds.

Results, in the legend of Figure 4 the authors comment on the results presented in Figure 1. This comment should be included in the text within the Results: … “Using antibodies kindly provided by Fabio Malavasi, University of Torino, Italy, CD38+ MCs were more frequently detected in the mammary gland (Figure 1d), and were also found in bone marrow and tonsils (Figure 1e–e′). At the same time, the highest freque-cy of immunohistochemical detection of CD38 was mirrors the high population of MCs in the skin (Figure 1f–f′,g,h,i–i′).”

A summary table of the results would be necessary, including the n of the observations made.

Author Response

To Reviewer 2

The authors' response to Comments and Suggestions for Authors is given below and highlighted italic:

The authors present an immunohistochemical evaluation of the cytological and histotopographic characteristics of CD38 expression in mast cells. The authors found that CD38 expression in a minority of the mast cell population. They indicate that CD38 expression is characterized by wide variability from low to high levels.

The authors are grateful to the Reviewer for a positive assessment of the work performed.

Results, in Table 3 must include the n corresponding to the number of assessment in each organ.

These comments of the Reviewer are taken into account in Table 3.  

Results, it is necessary that the figures maintain consistency with the symbols. The arrow must indicate the same type of result, and the double arrow a different result. The symbols should indicate the same type of observation in each of the figures. Moreover, for a better understanding of the results obtained, it would be necessary for the authors to indicate in each figure to which organ each image corresponds.

The authors thank the Reviewer for a thorough analysis of the figures in the article. The figures have been corrected accordingly. 

Results, in the legend of Figure 4 the authors comment on the results presented in Figure 1. This comment should be included in the text within the Results: … “Using antibodies kindly provided by Fabio Malavasi, University of Torino, Italy, CD38+ MCs were more frequently detected in the mammary gland (Figure 1d), and were also found in bone marrow and tonsils (Figure 1e–e′). At the same time, the highest frequency of immunohistochemical detection of CD38 was mirrors the high population of MCs in the skin (Figure 1f–f′,g,h,i–i′).”

The authors are grateful to the Reviewer for identifying the typo in the PDF file. The article contains corrections.

A summary table of the results would be necessary, including the n of the observations made.

These comments of the Reviewer are taken into account in Table 3.

The authors thank the Reviewer for useful comments.

Igor Buchwalow

Round 2

Reviewer 1 Report

Authors have appropriately responded to technical questions raised in my review and clarified that the main scope of this article is simply to show that CD38 can be immunohistochemically detected in paraffin sections. Unfortunately, the important biological and pathological meaning of CD38 expression in a small subset of mast cells was not addressed at all.

Author Response

To Reviewer 1

The authors' response to Comments and Suggestions for Authors is given below and highlighted italic:

Authors have appropriately responded to technical questions raised in my review and clarified that the main scope of this article is simply to show that CD38 can be immunohistochemically detected in paraffin sections. Unfortunately, the important biological and pathological meaning of CD38 expression in a small subset of mast cells was not addressed at all.

The authors are grateful to the Reviewer for the evaluation of the work performed and valuable suggestions on the analysis of CD38 expression in organ-specific populations of mast cells, which will significantly expand our understanding of the biology of mast cells. We will definitely take into account the reviewer's suggestions in future studies, and we will plan to devote a separate article to this issue.

The authors thank the Reviewer for useful comments.

Igor Buchwalow

Reviewer 2 Report

Figures still an issue. To improve the understanding the results of the manuscript all figures should maintain consistency with the symbols, and indicate in each figure to which organ each image corresponds. Not only in the legend.

In addition, figure legend should explain the results that are presented. In contrast,

In figure 4 panels, the authors indicate in panels (a–a′) “Mast cell co-localization with multiple CD38+ cells. The enlarged fragment (a′) shows the entry of tryptase-positive granules into the cytoplasm of CD38+ cells towards the nucleus (indicated by the arrow).”  Even though there is not any image, showing the CD38+ cells structure.

Likewise, when the authors indicate in figure 3, “(g–k) Tonsilla. Various variants of tryptase secretion and contact of mast cell granules with the plasmalemma of CD38+ cells (indicated by an arrow). “.

It will be important to know if there is any biochemical data to support that conclusion. Moreover, authors do not provide enough information from the image to conclude about differences in tryptase secretion. Authors should explain the results in the figure legends.

Similarly in figure 4 (b,c–c′,d).  Authors indicate “Contact of a mast cell with CD38+ cells, various variants of co-localization of tryptase-containing granules with the plasmalemma of CD38+ cells (indicated by an arrow)”.Authors do not provide enough information to demonstrate various variants of co-localization.

Author Response

To Reviewer 2

The authors' response to Comments and Suggestions for Authors is given below and highlighted italic:

Figures still an issue. To improve the understanding the results of the manuscript all figures should maintain consistency with the symbols, and indicate in each figure to which organ each image corresponds. Not only in the legend.

The authors thank the referee for the remark made. In accordance with the recommendations of the Reviewer, the names of the organs were added to the images of Figure 1, Figure 2 and Figure 3, in which photographs from various organs were presented. At the same time, Figure 4 remains unchanged, since only one organ (tonsilla) is represented here, which is reflected in the title: “Figure 4. Histoarchitectonics of specialized immune contacts of mast cells with CD38 + cells in tonsilla.”

In addition, figure legend should explain the results that are presented. In contrast,

In figure 4 panels, the authors indicate in panels (a–a′) “Mast cell co-localization with multiple CD38+ cells. The enlarged fragment (a′) shows the entry of tryptase-positive granules into the cytoplasm of CD38+ cells towards the nucleus (indicated by the arrow).” Even though there is not any image, showing the CD38+ cells structure. Likewise, when the authors indicate in figure 3, “(g–k) Tonsilla. Various variants of tryptase secretion and contact of mast cell granules with the plasmalemma of CD38+ cells (indicated by an arrow).“

The authors are grateful to the referee for a valuable comment on improving the interpretation of the photographs of the article. One of the main tasks of the work was not only a thorough analysis of CD38 expression in mast cells, but also visualization of the possible interaction of mast cells and other cells with CD38 expression. Since in the literature we did not find any significant information on the visualization of the interaction of mast cells and CD38+ cells, the data presented in our article show the fundamental possibility of studying in this direction using immunohistochemical protocols. At this point, the authors thank the Reviewer for highlighting a wide perspective for studying the various components of mast cell secretome on CD38+ cells. This might be the basis of a separate independent study.

It will be important to know if there is any biochemical data to support that conclusion.

With regard to this note of the Reviewer, we can confer that a lot of biochemical studies have been done in which attention has been paid to the formation of the enzymatic activity of tryptase, including intracellular and intragranular processing. In particular, comprehensive information is to this point is given in the review: “Atiakshin D, Buchwalow I, Samoilova V, Tiemann M. Tryptase as a polyfunctional component of mast cells. Histochem Cell Biol. 2018 May; 149 (5): 461-477".

Moreover, authors do not provide enough information from the image to conclude about differences in tryptase secretion. Authors should explain the results in the figure legends. Similarly in figure 4 (b,c–c′,d).  Authors indicate “Contact of a mast cell with CD38+ cells, various variants of co-localization of tryptase-containing granules with the plasmalemma of CD38+ cells (indicated by an arrow)”. Authors do not provide enough information to demonstrate various variants of co-localization.

The authors are grateful to the Reviewer for the comments made, emphasizing the importance, significance, and broad possibilities of interpreting the data obtained as a result of the implementation of the protocols of multiple immunomarking of multiplex immunohistochemistry. Within the framework of this publication, the authors' plans did not include the elucidation of organ-specific differences in tryptase secretion, including pathological conditions. The authors sought to show the fundamental possibilities of detecting the effects of tryptase on CD38-positive cells. This problem is quite extensive and deserves special attention in future studies. At this point, the authors would like to thank the Reviewer once again for presenting the future perspective studies on this topic. We believe that the value of detecting tryptase co-localization in granules and CD38-positive cells on histological sections, as well as the formation of specialized immune contacts of mast cells with CD38+ cells, is very important, which can significantly expand the interpretation window of morphological analysis of oncogenesis.

Micrographs providing morphological evidence of the regulatory effects of tryptase on CD38 + cells are discussed in Section 3.3: “Histotopographic Features of Co-Localization of Mast Cells and CD38 + Cells in a Specific Tissue Microenvironment”, as well as in Section 3.4: “Content of Tryptase in Nuclei of CD38 + Cells ". The authors give some interpretations of morphological pictures, in particular, “From the point of view of explaining the mechanism of tryptase entry into the nuclei of CD38 + cells, we can assume the regulatory consequences of this phenomenon. In particular, this phenomenon can affect the state of histones and cell proliferation, while tryptase exhibits anti-proliferative effects [27, 28]." However, based on the spectrum of the currently known biological activity of tryptase, localization of this protease within CD38+ cells can have much broader regulatory consequences that could be an issue for further studies.

The authors thank the Reviewer for useful comments.

Igor Buchwalow